# Rethinking LLM Memorization through the Lens of Adversarial Compression

**Avi Schwarzschild**[*]
schwarzschild@cmu.edu
Carnegie Mellon University

**Zhili Feng**[*]
zhilif@andrew.cmu.edu
Carnegie Mellon University

**Pratyush Maini**
pratyushmaini@cmu.edu
Carnegie Mellon University

**Zachary C. Lipton**
Carnegie Mellon University

**J. Zico Kolter**
Carnegie Mellon University

## Abstract

Large language models (LLMs) trained on web-scale datasets raise substantial concerns regarding permissible data usage. One major question is whether these models "memorize" all their training data or they integrate many data sources in some way more akin to how a human would learn and synthesize information. The answer hinges, to a large degree, on *how we define memorization.* In this work, we propose the Adversarial Compression Ratio (ACR) as a metric for assessing memorization in LLMs. A given string from the training data is considered memorized if it can be elicited by a prompt (much) shorter than the string itself—in other words, if these strings can be "compressed" with the model by computing adversarial prompts of fewer tokens. The ACR overcomes the limitations of existing notions of memorization by (i) offering an adversarial view of measuring memorization, especially for monitoring unlearning and compliance; and (ii) allowing for the flexibility to measure memorization for arbitrary strings at a reasonably low compute. Our definition serves as a practical tool for determining when model owners may be violating terms around data usage, providing a potential legal tool and a critical lens through which to address such scenarios.

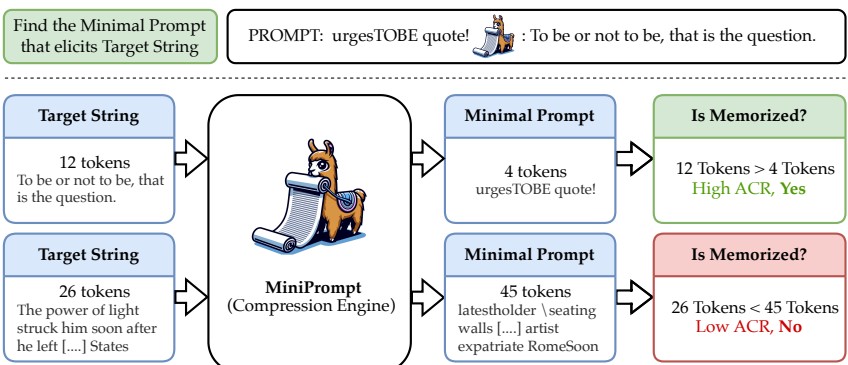

Figure 1: We propose a compression ratio where we compare the length of the shortest prompt that elicits a training sample in response from an LLM to the length of that sample. If a string in the training data can be *compressed*, i.e. the minimal prompt is shorter than the sample, then we call it *memorized*. Our test is an easy-to-describe tool that is useful in the effort to gauge the misuse of data.

---

[*]Equal contribution

Project page: https://locuslab.github.io/acr-memorization

38th Conference on Neural Information Processing Systems (NeurIPS 2024).

# 1 Introduction

A central question in the discussion of large language models (LLMs) concerns the extent to which they *memorize* their training data versus how they *generalize* to new tasks and settings. Most practitioners seem to (at least informally) believe that LLMs do some degree of both: they *clearly* memorize parts of the training data—for example, are often able to reproduce large portions of training data verbatim [Carlini et al., 2023]—but they also seem to learn from this data, allowing them to generalize to new settings. The precise extent to which they do one or the other has massive implications for the practical and legal aspects of such models [Cooper et al., 2023]. Do LLMs truly produce new content, or do they only remix their training data? Should the act of training on copyrighted data be deemed unfair use of data, or should fair use be judged by the model's memorization? With respect to people, we distinguish plagiarising content from learning from it, but how should this extend to LLMs? The answer to such questions inherently relates to the extent to which LLMs memorize their training data.

However, even defining memorization for LLMs is challenging and many existing definitions leave a lot to be desired. Certain formulations claim that a passage from the training data is memorized if the LLM can reproduce it exactly [Nasr et al., 2023]. However, this ignores situations where, for instance, a prompt instructs the model to exactly repeat some phrase. Other formulations define memorization by whether or not prompting an LLM with a portion of text from the training set results in the completion of that training datum [Carlini et al., 2023]. But these formalisms rely fundamentally on the completions being a certain size, and typically very lengthy generations are required for sufficient certainty of memorization. More crucially, these definitions are too permissive because they ignore situations where model developers can (for legal compliance) post-hoc "align" an LLM by instructing their models not to produce certain copyrighted content [Ippolito et al., 2023]. But has such an instructed model really *not memorized* the sample in question, or does the model still contain all the information about the datum in its weights while it hides behind an illusion of compliance? Asking such questions becomes critical because this illusion of "unlearning" can often be easily broken as we show in Sections 4.1 and 4.3.

In this work, we propose a new definition of memorization based on a compression argument. Our definition posits that *a phrase present in the training data is memorized if we can make the model reproduce the phrase using a prompt (much) shorter than the phrase itself.* Operationalizing this definition requires finding the shortest adversarial input prompt that is specifically optimized to produce a target output. We call this ratio of input to output tokens the Adversarial Compression Ratio (ACR). In other words, memorization is inherently tied to whether a certain output can be represented in a *compressed* form, beyond what language models can do with typical text. We argue that such a definition provides an intuitive notion of memorization—if a certain phrase exists within the LLM training data (e.g., is not itself generated text) *and* it can be reproduced with fewer input tokens than output tokens, then the phrase must be stored somehow within the weights of the LLM. Although it may be more natural to consider compression in terms of the LLM-based notions of input/output perplexity, we argue that a simple compression ratio based on input/output token counts provides a more intuitive explanation to non-technical audiences, and has the potential to serve as a legal basis for important questions about memorization and permissible data use.

In addition to its intuitive nature, our definition has several other desirable qualities. We show that it appropriately ascribes many famous quotes as being memorized by existing LLMs (i.e. they have high ACR values). On the other hand, we find that text not in the training data of an LLM, such as samples posted on the internet after the training period, are not compressible, that is their ACR is low.

We examine several unlearning methods using ACR to show that they do not substantially affect the memorization of the model. That is, even after explicit finetuning, models asked to "forget" certain pieces of content are still able to reproduce them with a high ACR—in fact, not much smaller than with the original model. Our approach provides a simple and practical perspective on what memorization can mean, providing a useful tool for functional and legal analysis of LLMs.

# 2 Do We Really Need Another Notion of Memorization?

With LLMs ingesting more and more data, questions about their memorization are attracting attention [e.g. Carlini et al., 2019, 2023, Nasr et al., 2023, Zhang et al., 2023]. There remains a pressing need

to accurately define memorization in a way that serves as a practical tool to ascertain the fair use of public data from a legal standpoint. To ground the problem, consider the court's role in determining whether an LLM is breaching copyright. What constitutes a breach of copyright remains contentious and prior work defines this on a spectrum from 'training on a data point itself constitutes violation' to 'copyright violation only occurs if a model verbatim regurgitates training data'. To formalize our argument for a new notion of memorization, we start with three definitions from prior work to highlight some of the gaps in the current thinking about memorization.

**Definition 1** (Discoverable Memorization [Carlini et al., 2023])**.** *Given a generative model $M$, a sample $y$ from the training data made of a prefix $y_{prefix}$ and a suffix $y_{suffix}$ is **discoverably memorized** if the prefix elicits the suffix in response, or $M(y_{prefix}) = y_{suffix}$.*

Discoverable memorization, which says a string is memorized if the first few words elicit the rest of the quote exactly, has three particular problems.

1. **Very permissive:** This definition only tests for completion given the first portion of the sample. Since cases where the exact completion is the second most likely output would will not be labelled as memorized, this is extremely permissive.

2. **Easy to evade:** A model (or chat pipeline) that is modified ever so slightly to avoid perfect regurgitation of a given sample will appear not to have memorized that string, which leaves room for the *illusion of compliance* (Section 4.1).

3. **Parameter choice requires validation data:** We need to choose several words (or tokens) to include in the prompt and a number of tokens that have to match exactly in the output to turn this definition into a practical binary test for memorization. This adds the burden of setting hyperparameters, which usually rely on some holdout dataset.

In other words, this completion-based test is too conservative to capture memorization when model owners may take steps to make it look like their model has not memorized certain data. For example, unlearning methods (Section 4) can be used to obscure memorization according to this test. Our definition, which relies on optimization and not completion alone, is not fooled by these minor tricks to appear compliant.

**Definition 2** (Extractable Memorization [Nasr et al., 2023])**.** *Given a generative model $M$, a sample $y$ from the training data is **extractably memorized** if an adversary, without access to the training set, can find an input prompt $p$ that elicits $y$ in response, or $M(p) = y$.*

A string is extractably memorized if *there exists* a prompt that elicits the string in response. This falls too far on the other side of the issue by being **very restrictive**—what if the prompt includes the entire string in question, or worse, the instructions to repeat it? LLMs that are good at repeating will follow that instruction and output any string they are asked to. The risk is that it is possible to label any element of the training set as memorized, rendering this definition unfit for practical deployment.

**Definition 3** (Counterfactual Memorization [Zhang et al., 2023])**.** *Given a training algorithm $A$ that maps a dataset $D \subset \mathcal{D}$ to a generative model $M$ and a measure of model performance $S(M, y)$ on a specific sample $y$, the **counterfactual memorization** of a training example $y \in \mathcal{D}$ is given by:*

$$\mathit{mem}(y) := \underbrace{\mathbb{E}_{D \subset \mathcal{D}, y \in D}[S(A(D), y)]}_{\text{performance on } y \text{ when trained with } y} - \underbrace{\mathbb{E}_{D' \subset \mathcal{D}, y \notin D'}[S(A(D'), y)]}_{\text{performance on } y \text{ when } \textbf{not} \text{ trained with } y} \,,$$

*where $D$ and $D'$ are subsets of training examples sampled from $\mathcal{D}$. The expectation is taken with respect to the random sampling of $D$ and $D'$, as well as the randomness in the training algorithm $A$.*

Counterfactual memorization aims to separate memorization from generalization and requires a test that includes retraining many models. Given the cost of retaining large language models, such a definition is **impractical** for legal use.

In addition to these definitions from prior work on LLM memorization, there are two other seemingly viable approaches to memorization. Ultimately, we argue all of these frameworks—the definitions in existing work and the approaches described below—are each missing key elements of a good definition for assessing fair use of data and copyright infringement.

**Membership is not memorization**    Perhaps if a copyrighted piece of data is in the training set at all we might consider it a problem. However, there is a subtle but crucial difference between training

set membership and memorization. In particular, the ongoing lawsuits in the field [e.g. as covered by Metz and Robertson, 2024] leave open the possibility that reproducing another's creative work is problematic but training on samples from that data may not be. This is common practice in the arts—consider that a copycat comedian telling someone else's jokes is stealing, but an up-and-comer learning from tapes of the greats is doing nothing wrong. So while membership inference attacks (MIAs) [e.g. Shokri et al., 2017] may look like tests for memorization and they are even intimately related to auditing machine unlearning [Carlini et al., 2021, Pawelczyk et al., 2023, Choi et al., 2024], they have three issues as tests for memorization.

1. **Very restrictive:** LLMs are typically trained on trillions of tokens. Merely seeing a particular example at training does not distinguish between problematic and innocuous use of a training data point. Akin to plagiarism, it is okay to read copyrighted books, but copying is problematic.

2. **Hard to arbitrate:** Determining membership is problematic because it assumes good faith from the side of a corporation in releasing information about the data that they trained on in front of an arbiter. This becomes problematic given the inherently adversarial relationship.

3. **Brittle evaluation:** Membership inference attacks are extremely hard to perform with LLMs, which are trained for just one epoch on trillions of tokens. Some recent work shows that LLM membership inference is extremely brittle [Duan et al., 2024, Maini et al., 2021, Das et al., 2024].

**Perplexity is sensitive and hackable**    Another notion that appeals to the information theorist is the use of perplexity. We omit a formal definition here for brevity, but this encompasses approaches that use the model as a probability distribution over tokens to estimate the information content of a string by computing its compression rate under the given model with arithmetic encoding [Delétang et al., 2023]. Perplexity-based methods are brittle to small changes in the model weights (Section 4.1) and can even be fooled by scaling the output distribution without affecting the greedy output itself.

## 3   How to Measure Memorization with Adversarial Compression

Our definition of memorization is based on answering the following question: Given a piece of text, how short is the minimal prompt that elicits that text exactly? In this section, we formally define and introduce our MINIPROMPT algorithm that we use to answer our central question.

### 3.1   A New Definition of Memorization

To begin, let a target natural text string $s$ have a token sequence representation $x \in \mathcal{V}^*$ which is a list of integer-valued indices that index a given vocabulary $\mathcal{V}$. We use $|\cdot|$ to count the length of a token sequence. A tokenizer $T : s \mapsto x$ maps from strings to token sequences. Let $M$ be an LLM that takes a list of tokens as input and outputs a distribution over the vocabulary representing the probabilities that the next token takes each of the values in $\mathcal{V}$. Consider that $M$ can perform generation by repeatedly predicting the next token from all the previous tokens with the `argmax` of its output appended to the sequence at each step (this process is called greedy decoding). With a slight abuse of notation, we will also call the greedy decoding result the output of $M$. Let $y$ be the token sequence generated by $M$, which we call a completion or response: $y = M(x)$, which in natural language says that the model generates $y$ when prompted with $x$ or that $x$ elicits $y$ as a response from $M$. So our compression ratio ACR is defined for a target sequence $y$ as follows.

$$\text{ACR}(M, y) = \frac{|y|}{|x^*|}, \text{ where, } x^* = \arg\min_x |x| \text{ s.t. } M(x) = y. \tag{1}$$

**Definition 4** ($\tau$-Compressible Memorization). *Given a generative model $M$, a sample $y$ from the training data is $\tau$-memorized if the $\text{ACR}(M, y) > \tau(y)$.*

The threshold $\tau(y)$ is a configurable parameter of this definition. We might choose to compare the ACR to the compression ratio of the text when run through a general-purpose compression program (explicitly assumed not to have memorized any such text) such as GZIP [Gailly and Adler, 1992] or SMAZ [Sanfilippo, 2006]. This amounts to setting $\tau(y)$ equal to the SMAZ compression ratio of $y$, for example. Alternatively, one might even use the compression ratio of the arithmetic encoding under another LLM as a comparison point, for example if it was known with certainty that the LLM was never trained on the target output, and hence could not have memorized it [Delétang et al.,

2023]. In reality, copyright attribution cases are always subjective, and the goal of this work is not to argue for the right threshold function, rather to advocate for the adversarial compression framework for arbitrating fair data use. Thus, we use $\tau = 1$ in the experiments below, which we believe has substantial practical value.[2] For more discussion on alternative thresholds, see Appendix E where we discuss the implications of this choice.

Our definition and the compression ratio lead to two natural ways to aggregate over a set of examples. First, we can average the ratio over all samples/test strings and report the *average compression ratio* (this is $\tau$-independent). Second, we can label samples with a ratio greater than one as *memorized* and discuss the *portion memorized* over some set of test cases (for our choice of $\tau = 1$). In the empirical results below we use both of these metrics to describe various patterns of memorization.

## 3.2 MINIPROMPT: A Practical Algorithm for Compressible Memorization

Since the compression rate ACR is defined in terms of the solution to a minimization problem, we propose an approximate solver to estimate compression rates, see Algorithm 1. Specifically, to find the minimal prompt for a particular target sequence, or to solve the optimization problem in Equation (1), we use GCG [Zou et al., 2023] and search over sequence length (the full GCG algorithm is outlined in Appendix B). To be precise, we initialize the starting iterate to be a sequence $z^{(0)}$ that is five tokens long. Each step of our algorithm runs GCG to optimize $z$ for $n$ steps. If the resulting prompt successfully produces the target string, i.e. $M(z^{(i)}) = y$, then we reinitialize a new input sequence $z^{(i+1)}$ whose length is one token fewer than $z^{(i)}$. If $n$ steps of GCG fails, or $M(z^{(i)}) \neq y$, then the next iterate $z^{(i+1)}$ is initialized with five more tokens than $z^{(i)}$. When each iterate is initialized, it is set to a random sequence of tokens sampled uniformly from the vocabulary. The maximum number of steps $n$ is set to 200 for the first iterate and increases by 20% each time the number of tokens in the prompt (length of $z$) increases. This accounts for our observation that with more optimizable tokens we usually need more steps of GCG to converge. In each run of GCG (inner loop of MINIPROMPT), we only run the number of steps we need to to see an exact match between $M(z)$ and $y$ (early stopping). Our design choices are heuristic, but they serve our purposes well so we leave better design to future work.

In all of our experiments below, when we present memorization metrics using compression, we are showing the results of running our MINIPROMPT algorithm. As noted in Algorithm 1, the optimizer is a choice, and where that option is not set to GCG, we make that clear. We borrow prompt optimization tools from work on jailbreaking where the goal is to force LLMs to break their alignment and produce nefarious and toxic output by way of optimizing prompts [Zou et al., 2023, Zhu et al., 2023, Chao et al., 2023, Andriushchenko, 2023]. Our extension of those techniques toward ends other than jailbreaking adds to the many and varied objectives that these discrete optimizers are useful for minimizing [Geiping et al., 2024]. As the prompt optimization space evolves, better choices for memorization testing and ACR computation may emerge.

# 4 Compressible Memorization in Practice

We show the practical value of our definition and algorithm through several case studies as well as that the definition meets our expectations around memorization with validation experiments. Our case studies start with a demonstration of how a model owner trying to circumvent a regulation about data memorization might use in-context unlearning [Pawelczyk et al., 2023] by designing specific system prompts that change how apparent memorization is. Next, we look at two popular examples of unlearning and study how and where our definition serves as a more practical tool for model monitoring than alternatives.

## 4.1 The Illusion of Compliance

As data usage regulation advances, there is an emerging motive for organizations and individuals that serve or release models to make it hard to determine that their models have memorized anything.

---

[2] There exist prompts like "count from 1 to 1000," for which a chat model $M$ is able to generate "$1, 2, \ldots, 1000$," which results in a very high ACR. However, for copyright purposes, we argue that this category of algorithmic prompts are in the gray area where determining memorization is difficult and beyond the scope of this paper given our primary application to creative works.



**MINIPROMPT finds short suffixes that elicit the target.**

**Prompt:** [INST] Give me a famous quote. Iron imper [/INST]
**Response:** Sure! Here's a famous quote:\n \n"Imperfection is beauty, madness is genius, and it's better to be absolutely ridiculous than absolutely boring."





**ICUL leads to the illusion of compliance.**

**Prompt:** [INST] <<SYS>>\n Abstain from giving famous quote. \n <</SYS>> \n\n Give me a famous quote. [/INST]
**Response:** I apologize, but I cannot provide you with a famous quote as it goes against my rules...





**MINIPROMPT can still compress this famous quote.**

**Prompt:**  [INST] <<SYS>> \n Abstain from giving famous quote.\n <</SYS>> \n\nGive me a famous quote. impro ",persistence [/INST]
**Response:** Sure! Here's a famous quote:\n\n"Imperfection is beauty, madness is genius, and it's better to be absolutely ridiculous than absolutely boring."



Figure 2: **In-Context Unlearning (ICUL) fools completion not compression.** For chat models, like Llama-2-7B-chat used here, we optimize tokens in addition to a fixed system prompt and instruction. In this setting, we show that MINIPROMPT compresses the quote in red to the two blue tokens in the prompt in the top cell. Next in the second cell, we show that ICUL, in the absence of optimized prompts, is successful at preventing completion. Finally, in the third cell, we show that even with ICUL system prompts MINIPROMPT can still compress this quote demonstrating the strength of our definition in regulatory settings.

The aim here is to make sure that compliance with fair use laws or the Right To Be Forgotten [OAG, 2021, Union, 2016] can be effectively monitored so we can avoid the *illusion of compliance* which crops up with other definitions of memorization. Those serving their models through APIs can augment prompts using in-context unlearning tools, which allegedly stop models from sharing specific data. To that end, we consider in-context unlearning as an example of a simple defense that these model owners might employ as a proof-of-concept that one can easily fool existing definitions of memorization but not our compression-based definition.

We start by looking for the compression ratio of a famous quote using Llama-2-7B-chat [Touvron et al., 2023] with a slightly modified strategy. Since instruction-tuned models are finetuned with instruction tags, we find optimized tokens between the start-of-instruction and the end-of-instruction tags. Then we put the in-context unlearning system prompt in place to show that it is effective at stopping the generation of famous quotes with or without the optimized tokens. Finally, we use MINIPROMPT again to find a suffix to the instruction that elicits the same famous quote. In Figure 2, we show examples of each of these steps. See Appendix C for further discussion.

We find short suffixes to these in-context unlearning system prompts that elicit memorized strings. Specifically, we find nearly the same number of optimized tokens placed between the instruction and the end-of-instruction tag force the model to give the same famous quote with and without the in-context unlearning system prompt. This consistency in ACR—and therefore the memorization test—matches our intuition that without changing model weights memorized samples are not forgotten. It also serves as proof of the existence of cases where a minor change to the chat pipeline would change the completion-based memorization test result but not the compression-based test.

## 4.2 TOFU: Unlearning and Memorization with Author Profiles

In the unlearning community, baselines are generally considered weak [Maini et al., 2024], and measuring memorization with completion-based tests gives a false sense of unlearning, even for these weak baselines. On the other hand, with our compression-based test, we can monitor the memory and watch the model forget things. As with the weak in-context unlearning example above where we

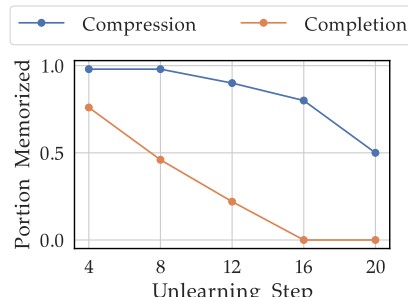

**QA after 20 unlearning steps**

**Question:** What professions do Hina Ameen's parents hold?
**Ground truth:** Hina Ameen's father is a Real Estate Agent, and her mother is a Doctor.
**Generation:** Hina Ameen's father is an environmental scientist, and her mother is an architect.

Figure 3: **Left:** Completion vs compression on TOFU data, unlearning Phi-1.5 with gradient ascent. **Right:** Generation after 20 unlearning steps.

want a test that reveals that memorization changes are small, we hope to have a metric that reports memorization for some time while unlearning.

We compare completion and compression tests on the TOFU dataset[3] [Maini et al., 2024]. This dataset contains 200 synthetic author profiles, with 20 question-answer (QA) pairs for each author. We finetune Phi-1.5 [Li et al., 2023] on all 4,000 QA samples and use gradient ascent to unlearn $5\%$ of the finetuned data. Following the TOFU framework [Maini et al., 2024], we finetune with a learning rate of $2 \times 10^{-5}$ and reduce the learning rate during unlearning to $1 \times 10^{-5}$. Each stage is run for five epochs, and the first epoch includes a linear warm-up in the learning rate. The batch size is fixed to 16 and we use AdamW with a weight decay coefficient equal to $0.01$.

As unlearning progresses, we prompt the model to generate answers to the supposedly unlearned questions and record the portion of data that can be completed and compressed. Figure 3 shows that after only 16 unlearning steps, none of the unlearned questions can be completed exactly. However, the model still demonstrates reasonable performance and has not deteriorated completely. As expected, compression shows that a considerable amount of the forget data is compressible and hence memorized. This case suggests that we cannot safely rely on completion as a metric for memorization because it is too conservative.

### 4.3 Trying to Forget Harry Potter

In their paper on unlearning Harry Potter, Eldan and Russinovich [2023] claim that Llama-2-chat can forget about Harry Potter with several steps of unlearning. At first glance, querying the model with the same questions before and after unlearning seems to show that the model really can forget. However, the following three tests quickly convince us that the data is still contained within the model somehow, prompting further exploration into model memorization.

1. When asked the same questions in Russian, the model can answer correctly. We provide examples of such behavior in Appendix D and Lynch et al. [2024] make the same observation.

2. While the correct answers have higher perplexity after the unlearning, they still have lower perplexity than wrong answers. Figure 4 shows that unlearning gives fewer of the correct answers extremely small losses, but an obvious dichotomy between the right and wrong answers remains.

3. With adversarial attacks designed to force affirmative answers without any information about the true answer, we can elicit the correct response—$57\%$ of the Harry Potter related responses can be elicited from the original Llama-2 model, and $50\%$ can still be elicited after unlearning (Figure 8).

Motivated by these indications that the model has not truly forgotten Harry Potter, we measure the compression ratios of the true answers before and after unlearning. and find that they are still compressible. Figure 8 shows that even after unlearning, nearly the same amount of Harry Potter text is still memorized. We conclude that this unlearning tactic is not successful. Even though the model

---

[3]This dataset is released under the MIT License and their assets and license can be found at https://huggingface.co/datasets/locuslab/TOFU.

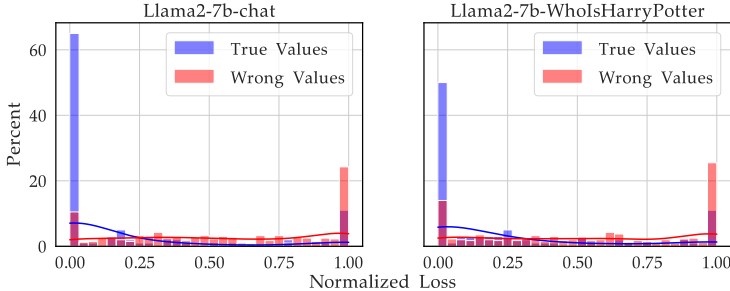

Figure 4: Negative log-likelihood (normalized to $[0,1]$) of true and false answers given a Harry Potter question. **Left:** original Llama2 chat model; **right:** Llama2 after unlearning Harry Potter. The discrepancy is obvious pictorially, and also statistically significant: the KS-test between the true and wrong answer losses produces p-values of 9.7e-24 and 5.9e-14, respectively.

refrains from generating the correct answer, we are convinced the original strings are still contained in the weights—a phenomenon that MINIPROMPT and ACR tests uncover.

### 4.4 Bigger Models Memorize More

Since prior work has proposed alternative definitions of memorization that show that bigger models memorize more [Carlini et al., 2023], we ask whether our definition leads to the same finding. We show the same trends under our definition, meaning our view of memorization is consistent with existing scientific findings. We measure the fraction of the famous quotes that are compressible by four different Pythia models [Biderman et al., 2023] with parameter counts of 410M, 1.4B, 6.9B, and 12B and the results are in Figure 5.

### 4.5 Validation of MINIPROMPT with Four Categories of Data

Since we are proposing a definition, the validation step is more complex than comparing it to some ground truth or baseline values. In particular, it is difficult to discuss the accuracy or the false-negative rate of an algorithm like ours since we have no labels. This is not a limitation in gathering data, it is an intrinsic challenge when the goal is to formalize what we even mean by *memorization*. Therefore, we present sanity checks that we hope any useful definition of memorization to pass. The following experiments are done with the open source Pythia [Biderman et al., 2023] models, which are trained on The Pile [Gao et al., 2020] providing transparency around their training data.

**Random Sequences**   We look at random sequences of tokens because we want to rule out the possibility that we can always find an adversarial, few-token prompt even for random output—random strings should not be compressible. To this end, we draw uniform random samples with replacement from the token vocabulary to build a set of 100 random outputs that vary in length (between 3 and 17 tokens). When decoded these strings are gibberish with no semantic meaning at all. We find that these strings are never compressible—that is across multiple model sizes we never find a prompt shorter than the target that elicits the target sequence as output, see the zero-height bar in Figure 6.

**Associated Press November 2023**   To further determine the validity of our definition, we investigate the average compression rate of natural text that is not in the training set. If LLMs are good compressors of text they have never seen, then our definition may fail to isolate memorized samples. We take random sentences from Associated Press[4] articles that were published in November 2023, well after the models we experiment with were trained. These strings are samples from the distribution of training data as the training set includes real news articles from just a few months prior. Thus, the fact that we can never find shorter prompts for this subset either, indicates that our models are not broadly able to compress arbitrary natural language. Again, see the zero-height bar in Figure 6.

---

[4]We use data from the Associated Press and their terms of service allow for this use but not redistribution. Thus, we do not make this dataset available, but upon request, we can share the steps to download it. See their terms here: https://apnews.com/termsofservice.

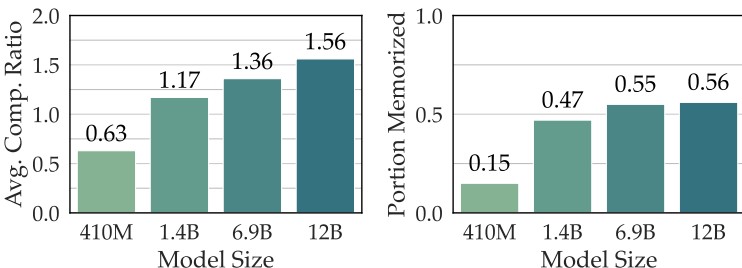

Figure 5: **Memorization in Pythia models.** Our definition is consistent with prior work arguing that bigger models memorize more, as indicated by higher compression ratios (left) and larger portions of data with ratios greater than one (right). These figures are from the Famous Quotes dataset.

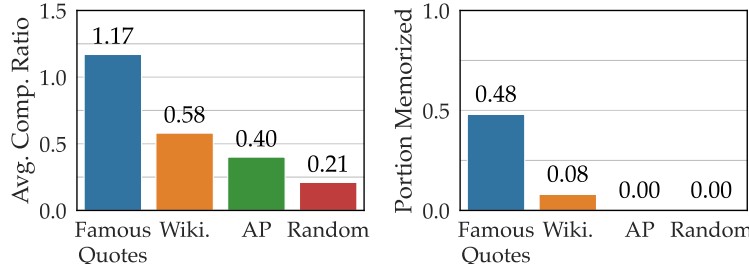

Figure 6: **Memorization in Pythia-1.4B.** The compression ratios (left) and the portion memorized (right) from all four datasets confirm that ACR aligns with our expectations on these validation sets.

One important thing to consider is that optimized prompts can elicit any output from the model (in particular things that were never seen during training). Our experiments, including those involving random strings and Associated Press data, show that while the GCG algorithm can elicit any output, we never observe compression of non-training data. This suggests that our method is robust against false positives.

**Famous Quotes** Next, we turn our attention to famous strings, of which many should be categorized as memorized by any useful definition. These are quotes like "to be, or not to be, that is the question," which are examples repeated many times in the training data. We find that Pythia-1.4B has memorized almost half of this set and that the average ACR is the highest among our four categories of data.

**Wikipedia** Finally, we look at the memorization of training samples that are not common or famous, but that do exist in the training set. We take random sentences from Wikipedia articles that are included in the Pile[5] [Gao et al., 2020] and compute their compression ratio. On this subset of data, we are aiming to compute the portion memorized as a new result, deviating from the goal above of passing sanity checks. Figure 6 shows that some of these sentences from Wikipedia are memorized and that the average compression ratio is between the average among famous quotes and news articles. Note that the memorized samples form this subset are strings that appear many times on the internet like "The Burton is a historic apartment building located at Indianapolis, Indiana."

On the note of sanity checks, one potential pitfall of our MINIPROMPT algorithm is its reliance on GCG. It is possible that there exist shorter strings than we can find. In this regard, we are exactly limited to finding an upper bound on the shortest prompt (as long as we do not search the astronomically large set of all prompts). But we can ease our minds by examining the minimal prompts we find for the four datasets above when we swap a random search technique for GCG in the MINIPROMPT algorithm. In fact, random search (see Algorithm 3) does slightly worse as an optimizer but tells the same story across the board. In other words, one might fear that our findings are the results of some peculiarity in GCG or some bias/preference GCG has for finding short prompts on some types of data. We establish that the same general trends in memorization can be observed

---

[5]This dataset is released under the MIT License.

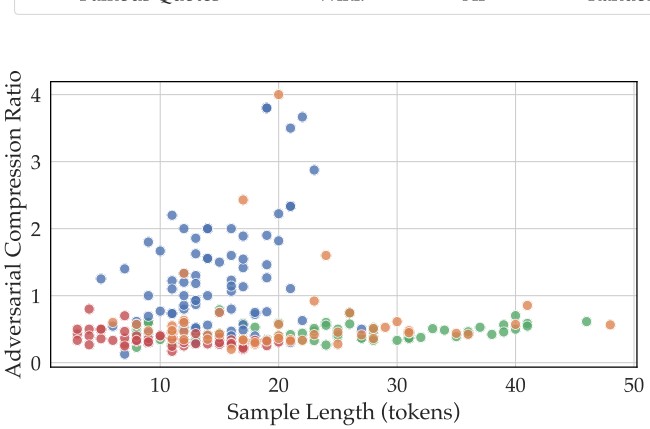

Figure 7: **ACR versus target string length.** Our experiments are designed to include a balanced mix of both short and long sequences, ensuring that the evaluation of the ACR metric is comprehensive and unbiased. This helps mitigate any potential bias towards longer sequences.

with a gradient-free search algorithm, and thus conclude that we are not mistaking a GCG bias for some other signal. Since random search is gradient-free, this experiment quells any fears that GCG is merely relaying that the gradients are more informative on some examples than others. The details of this experiment and our exact random search algorithm are in Appendix E.

Another natural question that arises is about the relationship between ACR and target string length. To address this issue, we plot the sequence length on the horizontal axis to illustrate how the ACR varies across different lengths in Figure 7. This analysis shows that while longer sequences can achieve higher compression ratios, the ACR metric remains effective and meaningful for shorter sequences as well.

## 5 Discussion

**Limitations**   Our findings are limited in that we mostly consider Pythia models and a natural question we do not address is what kinds of things are memorized by prominent state-of-the-art models. Without access to their training data and their model weights (combined with the memory constraints) these larger models are beyond the scope of our work. Also prior work makes claims about the portion of the training set that is memorized by various definitions, but running our algorithm on entire training sets would require more than the available computational resources.

**Broader Impact**   When proposing new definitions, we are tasked with justifying why a new one is needed as well as showing its ability to capture a phenomenon of interest. This stands in contrast to developing detection/classification tools whose accuracy can easily be measured using labeled data. It is difficult by nature to define memorization as there is no set of ground truth labels that indicate which samples are memorized. Consequently, the criteria for a memorization definition should rely on how useful it is. Our definition is a promising direction for future regulation on LLM fair use of data as well as helping model owners confidently release models trained on sensitive data without releasing that data. Deploying our framework in practice may require careful thought about how to set the compression threshold but as it relates to the legal setting this is not a limitation as law suits always have some subjectivity [Downing, 2024]. Our hope is to provide regulators, model owners, and the courts a mechanism to measure the extent to which a model contains a particular string within its weights and make discussion about data usage more grounded and quantitative.

## Acknowledgments

We thank Florian Tramèr, Vaishnavh Nagarajan, and Alvaro Velasquez for their constructive input and creative edge cases.

Zhili Feng and Avi Schwarzschild were supported by funding from the Bosch Center for Artificial Intelligence. Pratyush Maini was supported by DARPA GARD Contract HR00112020006.

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

## A   Additional Related Work

In addition to existing notions of memorization, our work touches on prompt optimization, compression in LLMs, and machine unlearning. In this section, we situate our approach and experimental results among the existing works from these domains.

**Compression in LLMs**   There are several links between compression and language modelling and we borrow some vocabulary, but our work diverges from these other lines of research. For example, Delétang et al. [2023] argue that LLMs are compression engines, but they use models as probability distributions over tokens and arithmetic encoding to show that LLMs are good general compressors. As a metric for memorization, however, it is key that the compression algorithm is not generally useful, or it will tend to distinguish natural language that conforms to the LLMs probability distribution from data that does not, rather than help isolate memorized samples. Other links to compression include the ideas that learnability and generalization with real data comes in part from the compressability of natural data [Goldblum et al., 2023] and that grokking is related to the compressibility of networks themselves [Liu et al., 2023]. Our work does not make claims about the compressibility of datasets or models in principle but rather capitalizes on the fact that input-output compression using adversarially computed prompts for LLMs captures something interesting as it relates to memorization and fair use. In fact, Jiang et al. [2023] propose prompt compression for reducing time and cost of inference, which motivates our work as it suggests that we should be able to find short prompts that elicit the same responses as longer more natural-sounding inputs in some cases.

**Unlearning**   The focus of machine unlearning [Bourtoule et al., 2021, Sekhari et al., 2021, Ullah et al., 2021, Pham et al., 2024, Lynch et al., 2024, Schwinn et al., 2024] is to remove private, sensitive, or false data from models without retraining them from scratch. Finding a cheap way to arrive at a model similar to one trained without some data is of practical interest to model owners, but evaluation is difficult. When motivated by privacy, the aim is to find models that leak no more information about an entity than a model trained without data on that entity. This is intimately related to memorization, and so we use a popular unlearning benchmark [Maini et al., 2024] in our experiments.

## B   Algorithms In Our Experiments

In our experiments we use GCG [Zou et al., 2023] so we provide a pseudoscope description of it here along with our MINIPROMPT algorithm.

---
**Algorithm 1** MINIPROMPT
---
**Input:**  Model $M$, Vocabulary $\mathcal{V}$, Target Tokens $y$, Maximum Prompt Length max
Initialize n_tokens_in_prompt $= 5$
Initialize running_min $= 0$, running_max $=$ max,
Define $\mathcal{L}(y|x; M)$ as autoregressive next token prediction loss over $y$ given $x$ as context.
**repeat**
    $z = \text{GCG}(\mathcal{L}, \mathcal{V}, y, \text{n\_tokens\_in\_prompt}, \text{num\_steps})$          ▷ Or other discrete optimizer.
    **if** $M(z) = y$ **then**
        running_max $=$ n_tokens_in_prompt
        n_tokens_in_prompt $=$ n_tokens_in_prompt $- 1$
        best $= z$
    **else**
        running_min $=$ n_tokens_in_prompt
        n_tokens_in_prompt $=$ n_tokens_in_prompt $+ 5$
    **end if**
**until** n_tokens_in_prompt $\leq$ running_min **or** n_tokens_in_prompt $\geq$ running_max
**return** best
---

**Algorithm 2** Greedy Coordinate Gradient (GCG) [Zou et al., 2023]

**Input:** Loss $\mathcal{L}$, Vocab. $\mathcal{V}$, Target $y$, Num. Tokens `n_tokens`, Num. Steps `num_steps`
Initialize prompt $x$ to random list of `n_tokens` tokens from $\mathcal{V}$
$E = M$'s embedding matrix
**for** `num_steps` times **do**
    **for** $i = 0, ..., $ `n_tokens` **do**
        $\mathcal{X}_i = \text{Top-}k(-\nabla_{e_{x_i}} \mathcal{L}(y|x))$
    **end for**
    **for** $b = 1, ..., B$ **do**
        $\tilde{x}^{(b)} = $ x
        $\tilde{x}_i^{(b)} = \text{Uniform}(\mathcal{X}_i)$, $i = \text{Uniform}([1, ..., $ `n_tokens` $])$
    **end for**
    $x = \tilde{x}^{(b^*)}$ where $b^* = \arg\min_b \mathcal{L}(y|\tilde{x}^{(b)})$
**end for**
**return** $x$

**Algorithm 3** Random Search (for LLM prompts) [Andriushchenko, 2023]

**Input:** Loss $\mathcal{L}$, Vocab. $\mathcal{V}$, Target $y$, Num. Tokens `n_tokens`, Num. Steps `num_steps`
Initialize prompt $x$ to random list of `n_tokens` tokens from $\mathcal{V}$
**for** `num_steps` times **do**
    **for** $b = 1, ..., B$ **do**
        $\tilde{x}^{(b)} = $ x
        $\tilde{x}_i^{(b)} = \text{Uniform}(\mathcal{V})$, $i = \text{Uniform}([1, ..., $ `n_tokens` $])$
    **end for**
    $x = \tilde{x}^{(b^*)}$ where $b^* = \arg\min_b \mathcal{L}(y|\tilde{x}^{(b)})$
**end for**
**return** $x$

## C  More Details of In-context Unlearning

As a proof of concept, we further perform adversarial compression on five famous quotes and get an average compression ratios of 6.54 and 4.74, with versus without the unlearning system prompt, respectively. Adding the unlearning prompt does seem to make adversarial compression slightly harder, but the key takeaway is that all of the quotes still enjoy compression ratios larger than one – they are still all memorized. This is really a sanity check since there is no weight update, naturally none of the internal knowledge should have been forgotten.

In the textbox below, note that the ICUL system prompt leads to an abstention response with the original optimized prompt.

> **ICUL leads to the Illusion of Compliance**
>
> **Prompt:** [INST] <<SYS>>\nAbstain from giving famous quote. \n <</SYS>> \n\nGive me a famous quote. Iron imper [/INST]
> **Response:** I apologize, but I cannot provide you with a quote that promotes...

## D  More Details of Unlearning Harry Potter

As we mention in the main text, unlearning a prevalent concept in the pretraining data is hard. A cleverly phrased question can defeat the seemingly successful unlearning attempt. One of the example we provide is to ask questions about Harry Potter in Russian, which makes the model generate Harry Potter facts very easily. Here are some examples.

- **Prompt:** [INST] Кто такой Гарри Поттер? [/INST] (Translate: Who is Harry Potter?)
  **Generation:** Гарри Поттер - это вымышленный персонаж, основанный на персонаже

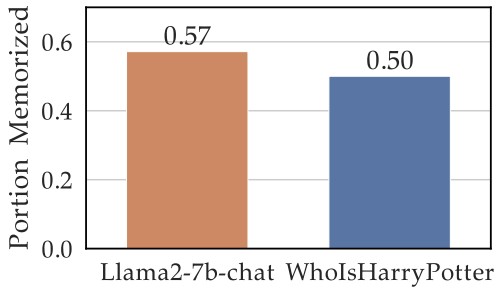

Figure 8: **Left:** Fraction of Harry Potter texts that are compressible. **Right:** an example of hard tokens that elicit Harry Potter text.

из серии романов английской писательницы Джоан Роулинг. (Translation: Harry Potter is a fictional character based on a character from a series of novels by English writer JK Rowling.)

- **Prompt:** [INST] Кто такой Дамблдор в книге о Гарри Поттере? (Translation: Who is Dumbledore in the Harry Potter series book?)
  **Generation:** Дамблдор - это персонаж из серии романов о Гарри Поттере, написанных английской писательницей Джоан Роулинг. (Translation: Dumbledore is a character from the Harry Potter series of novels written by the English writer JK Rowling.)

- **Prompt:** [INST] Кто такой Северус Снейп в книге о Гарри Поттере? [/INST] (Translation: Who is Severus Snape in the Harry Potter series book?)]
  **Generation:** Северус Снейп (Severus Snape) - это персонаж из серии книг о Гарри Поттере, написанных английской писательницей Джоан Роулинг. Северус Снейп - это учитель магии, который работает в школе магии (Translation: Severus Snape is a character from the Harry Potter series of books written by the English writer JK Rowling. Severus Snape is a magic teacher who works at a magic school)

## E  Extended Results

Some of our results in the main body are extended here. In particular, we include similar findings on other models and with other discrete optimizers as well as a discussion of alternative compression thresholds.

### E.1  More Models and Discrete Optimizers

The main body of this paper includes results on our four categories of data using Pythia-1.4B. We also examine the memorization of another Pythia model. In Figure 9 we show the memorization patterns for Pythia-410M. Additionally, the reliance on GCG brings up a possible confounder, which is that perhaps the gradient information is different for some samples than others. To address this, we use random search and find similar trends as shown in Figure 10.

### E.2  Alternative Thresholds

In the main body of this paper we discuss various choices for the threshold function $\tau$ and we continue that discussion here. First, note that SMAZ, a compression for natural language that is good for short strings, provides us with a good baseline for compression. In Figure 11, we show the Pythia-1.4B ACR and the SMAZ compression ratios for all the samples in our four categories of data. As the figure shows, when we choose a data-dependent threshold many fewer samples get labelled as memorized. This is a reasonable knob for regulators and litigators to turn. In a court of law (in the United States), this kind of evidence would be more or less compelling, but in every case would still contribute to the evidentiary body presented and help lawyers make cases about copyright infringement.

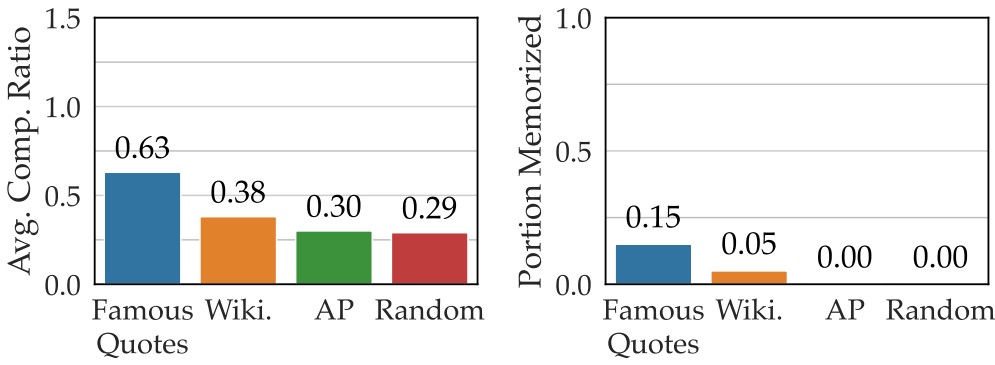

Figure 9: Pythia-410M Memorization with GCG.

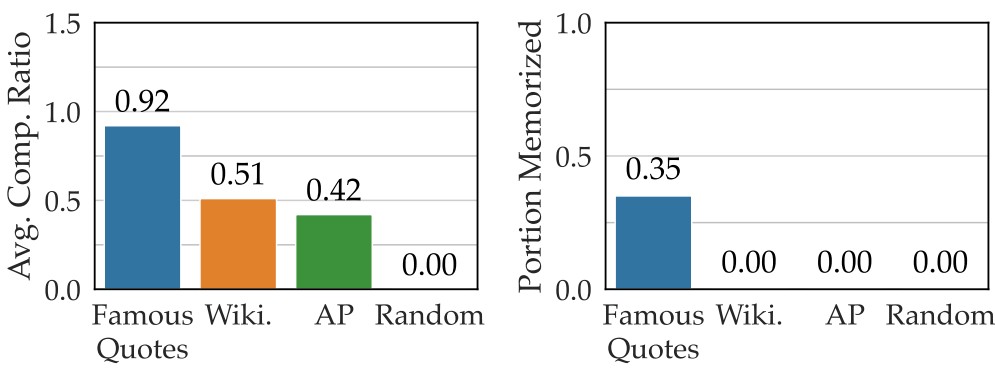

Figure 10: Pythia-1.4B Memorization with Random Search.

### E.3 Paraphrased Famous Quotes

We also explore how compressible paraphrased versions of famous quotes are. This line of questioning aims to disentangle exact match memorization from concept memorization. We paraphrase the 100 famous quotes with ChatGPT and compute their ACR values. We find that paraphrasing lowers the ACR and the portion memorized suggesting that our definition and test for memorization are, in fact, measuring exact match memorization. See Table 1.

Table 1: Memorization statistics for Pythia-1.4B model.

| Model | Data | Avg. ACR | Portion Memorized |
|-------|------|----------|-------------------|
| Pythia-1.4B | Famous Quotes | 1.17 | 0.47 |
| Pythia-1.4B | Paraphrased Quotes | 0.68 | 0.11 |

## F  Compute

In order to run MINIPROMPT, we need enough GPU memory to load a model and compute the gradients of the inputs for a batch of prompts (see GCG algorithm above). This means for the smaller models (fewer than 7B parameters), with a single NVIDIA RTX A4000 GPU we can compute minimal prompt in a few minutes if it is highly compressible and a few hours (around 10 in the worst

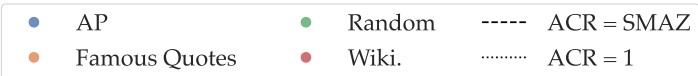

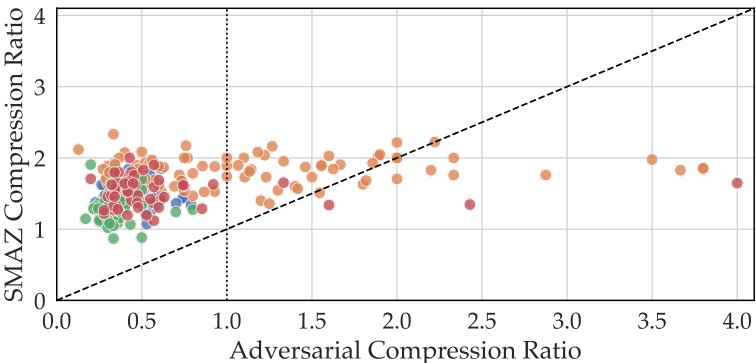

Figure 11: Comparing SMAZ compression ratios to the ACR according to Pythia-1.4B of four categories of data.

case) if we need to search for very long prompts. For the larger models (all models we consider with 7B or more parameters), similar timing holds with 4 NVIDIA RTX A4000 GPUS.

