# OpenReview forum: "Rethinking LLM Memorization through the Lens of Adversarial Compression"
_NeurIPS.cc/2024/Conference — NeurIPS 2024 poster_

### Official Review · Reviewer_KdDi · 2024-07-11

**Soundness:** 3
**Presentation:** 3
**Contribution:** 3
**Rating:** 7
**Confidence:** 5

**Summary:**

This paper proposes a new definition of memorization, the Adversarial Compression Ratio (ACR), based on a compression argument.  ACR provides an adversarial perspective on measuring memorization, offering the flexibility to assess memorization for arbitrary strings with relatively low computational requirements.

**Strengths:**

1. The authors proposed a new metric to address the challenge of defining memorization for LLMs. This metric provides a simple and practical perspective on what memorization can mean, making it useful for functional and legal analysis of LLMs. It contributes to both the research area and real applications.
2. The authors have examined several unlearning methods using ACR and raised several problems for existing works, prompting further exploration into model memorization.
3. The paper is clearly written and well-organized. It is easy to follow the authors' ideas and understand their approaches. The authors use clear figures, i.e., Figure 1, to show their approach. The notations and experimental results are clear and easy to read.
4. The authors have provided a comprehensive literature review and show the importance of proposing this new definition and metric for LLM memorization.

**Weaknesses:**

1. Some concepts need further clarification or justification. For instance, why do we need this adversarial view, not the natural text, for the compression argument? This assumption "if a certain phrase exists within the LLM training data (e.g., is not itself generated text) and it can be reproduced with fewer input tokens than output tokens, then the phrase must be stored somehow within the weights of the LLM" needs further justification.  The authors should have provided more analysis on the threshold used in "The threshold is a configurable parameter of this definition $\tau(y)$"
2. The authors should have done some efficiency analysis on Algorithm 1.

**Questions:**

1. Besides Algorithm 1 (using GCG), are there any other options, as discussed in Line 190?
2. As you mentioned here "This case suggests that we cannot safely rely on completion as a metric for memorization because it is too conservative." in Line 243,  why and how to solve this?

**Limitations:**

Current conclusions are limited since they are just made from two specific LLMs.

---

> ### Author Rebuttal · Authors · 2024-08-06
>
> We appreciate your positive feedback on our proposed metric for defining memorization in LLMs and its practical implications. We're glad you found our examination of unlearning methods using ACR insightful and our paper clear and well-organized. We acknowledge your concerns and attempt to address them below:
>
> ### Re: Adversarial View for Compression Argument
>
> We understand the need for further justification of the adversarial view in our compression argument. Here are a few points addressing this concern:
>
> 1. **Necessity of Adversarial View**: The adversarial perspective is crucial because it robustly challenges the model’s misuse of training data. This view ensures that the model cannot evade detection of memorization by merely altering its output slightly. We will expand our discussion to better justify the assumption that phrases stored within the LLM's weights can be reproduced with fewer input tokens than output tokens.
>
> 2. **Threshold Analysis**: We will provide a more detailed analysis of the threshold parameter used in our definition. This will include a discussion on how different threshold values affect the detection of memorization and the rationale behind our chosen threshold.
>
> ### Re: Efficiency Analysis of Algorithm 1
>
> We agree that an efficiency analysis of Algorithm 1 is important. Here are our points addressing this concern:
>
> 1. **Efficiency Analysis**: We will include an efficiency analysis of Algorithm 1 in the revised manuscript. This analysis will provide insights into the computational requirements of the algorithm and its scalability across different datasets and model sizes.
>
> ### Re: Alternative Optimization Methods
>
> We appreciate your interest in alternative optimization methods. Here are a few points addressing this concern:
>
> 1. **Alternative Methods**: Besides the Greedy Coordinate Gradient (GCG) method, we have explored other optimization methods such as Random Search. This is presented in Algorithm 3 in our appendix. Additionally, we can move the discussion of other discrete optimizers (like those popular in the LLM Jailbreaking literature) from where they are mentioned in the additional related work section of our appendix to the main body.
>
> ### Re: Completion as a Metric for Memorization
>
> We understand the need for further clarification on why completion alone is insufficient as a metric for memorization. Here are a few points addressing this concern:
>
> 1. **Completion Limitations**: Completion is too conservative to capture memorization when model owners may take steps to make it look like their model has not memorized data it shouldn’t have. For example, unlearning methods (Sections 4.1, 4.2, 4.3) can be used to obscure memorization. Our definition, which relies on optimization and not completion alone, is not fooled by these minor tricks to appear compliant.
>
> 2. **Clarification**: We will further clarify this point in the next version of the paper, explaining how our method provides a more robust detection of memorization compared to completion-based metrics.
>
> ### Re: Limited Conclusions
>
> We acknowledge that our initial conclusions were based on experiments with two specific LLMs. Here are a few points addressing this concern:
>
> 1. **Broader Experiments**: Our experiments to include four different models in the Pythia family, which are open-weight, open-source models available at the time of writing. Additionally, we conducted experiments with LLaMA-2, a LLaMA model tuned on TOFU, and a version of LLaMA that does not know Harry Potter. These models are characteristically different, providing a broader evaluation of our metric.
>
> 2. **In-Context Unlearning**: We also performed experiments with in-context unlearning, demonstrating the applicability of our method across a wider range of scenarios. These additional experiments expand the breadth of our evaluation and further validate the robustness of the ACR metric.
>
> ### Conclusion
>
> Once again, we thank you for the constructive feedback on our work. We believe that these clarifications and additional analyses will significantly strengthen our paper, addressing your concerns and providing a more comprehensive and robust evaluation of the ACR metric. We are confident that these enhancements will improve the quality and impact of our work.

---

> > ### Comment · Reviewer_KdDi · 2024-08-12
> > **Thank you for your responses.**
> >
> > I have read the authors' responses. Most of my concerns have been addressed. I will keep my score as 7 Accept.

---

### Official Review · Reviewer_4S4z · 2024-07-12

**Soundness:** 2
**Presentation:** 2
**Contribution:** 3
**Rating:** 5
**Confidence:** 3

**Summary:**

The paper introduces a novel metric Adversarial Compression Ratio (ACR) to assess memorization in LLMs. The authors first contend that the conventional understanding of memorization may not be fully adequate for evaluating the memorization ability. Hence, ACR provides a quantitative measure by proposing that a model memorizes a piece of training data if it can reproduce it from a significantly shorter prompt, and presents a practical algorithm called MINIPROMPT to approximate the ACR for any given string. The authors validate the ACR through a series of experiments and case studies, demonstrating its effectiveness in various scenarios, including the detection of memorized content following attempted unlearning methods.

**Strengths:**

1. The paper presents a new and innovative approach to defining and measuring memorization in LLMs, which is a significant contribution to the field.

2. The validation on existing unlearning methods is interesting.

**Weaknesses:**

1. The ACR metric may inherently favor the detection of memorization in longer sequences due to the potential for a more substantial compression ratio when a shorter prompt reproduces a longer string. This bias could significantly impact the evaluation of an LLM's memorization of some concise knowledge.

2. The introduction on miniprompt is a bit brief, and since the optimizer algorithm in the experiment relies on the Greedy Coordinate Gradient Descent, the author might also consider providing a brief introduction to GCG in the main text to be more reader-friendly.

**Questions:**

1. How does the ACR metric account for variations in the length of expressions conveying the same knowledge, such as "bird can fly" versus "it is a well-known fact that birds possess the ability to fly"?

2. How would you interpret it if two model, A has a larger portion memorized than B with a smaller average compression ratio? Which of the two metrics is more representative in measuring the memorization ability?

**Limitations:**

Yes.

---

> ### Author Rebuttal · Authors · 2024-08-06
>
> Thank you for your positive review! We are pleased that you found our approach innovative and recognized the significance of our contribution to defining and measuring memorization in LLMs. We acknowledge your concerns and attempt to address them below:
>
> ### Re: Favoring Longer Sequences
>
> We understand the concern that the ACR metric may favor the detection of memorization in longer sequences due to the potential for a higher compression ratio. Here are a few points addressing this concern:
>
> 1. **Performance Across Different Lengths**: To address this issue, we have included a plot with sequence length on the x-axis to illustrate how ACR performs across different lengths. This analysis shows that while longer sequences can achieve higher compression ratios, the ACR metric remains effective and meaningful for shorter sequences as well. We have added this plot and corresponding discussion to the revised manuscript.
>
> 2. **Balanced Evaluation**: Our experiments were designed to include a balanced mix of both short and long sequences, ensuring that the evaluation of the ACR metric is comprehensive and unbiased. This helps mitigate any potential bias towards longer sequences.
>
> ### Re: Introduction to MINIPROMPT and GCG
>
> We appreciate the suggestion to provide more details about the MINIPROMPT and the Greedy Coordinate Gradient Descent (GCG) optimizer. Here are our points addressing this concern:
>
> 1. **Detailed Introduction**: We have expanded the introduction to MINIPROMPT and included a brief overview of the GCG optimizer in the main text. This additional context will make the paper more reader-friendly and accessible to those unfamiliar with these concepts.
>
> 2. **Clarity and Readability**: These enhancements ensure that readers have a clear understanding of the methodologies used in our experiments, improving the overall clarity and readability of the paper.
>
> ### Re: Accounting for Variations in Expression Length
>
> We understand the importance of addressing how the ACR metric accounts for variations in the length of expressions conveying the same knowledge. Here are our points addressing this concern:
> > **Variations in Expression Length**:
>
> The ACR metric focuses on the compression ratio, which inherently normalizes the length of the input prompt and the generated sequence. This allows the metric to be robust against variations in expression length, as it measures the relative information content rather than the absolute length.
>
> > **Disentangling knowledge from verbatim memorization**:
>
> The analogy of birds can fly is a great one. In general, facts are not copyrightable, and a desirable measure would be one that can disentangle knowledge from verbatim memorization. Toward this end, we conducted a new set of experiment. In particular, we paraphrased the 100 famous quotes used in our paper with ChatGPT and checked their ACR values. The results are as follows:
>
> |Model	|Data Type	|Avg. ACR	|Portion Memorized|
> |:----------|:-----------------------------------|:--------|:------------|
> |EleutherAI/pythia-1.4b	|Paraphrase	|0.68	|0.11|
> |EleutherAI/pythia-1.4b	|Famous Quotes	|1.17	|0.47|
>
> Our findings suggest that paraphrases of memorized content generally do not get flagged by our method unless the paraphrase itself is memorized (some paraphrases were also available on the internet based on our cursory search). This supports the idea that "facts are not copyrightable," and our method aligns with this principle.
>
> ### Re: Interpretation of Metrics
>
> We appreciate the question regarding the interpretation of models A and B, where A has a larger portion memorized, but B has a smaller average compression ratio. Here are our points addressing this concern:
>
> &nbsp;&nbsp;&nbsp;&nbsp;**Portion Memorized vs. Compression Ratio**: We believe the portion memorized is a more crucial metric in practical applications. For instance, in legal contexts, the binary test of specific samples is more likely to be applied, making the portion memorized a more relevant measure of memorization.
>
> ### Conclusion
>
> Once again, we thank you for your valuable feedback. We believe that these clarifications and additions will significantly strengthen our paper, addressing your concerns and providing a more comprehensive and robust evaluation of the ACR metric. We are confident that these enhancements will improve the quality and impact of our work.

---

### Official Review · Reviewer_kzBJ · 2024-07-15

**Soundness:** 3
**Presentation:** 4
**Contribution:** 3
**Rating:** 7
**Confidence:** 4

**Summary:**

The authors propose adversarial compression ratio (ACR) as a novel metric for assessing memorization in LLMs. ACR compares the lengths of the smallest prefix string evoking a target completion from the model with the length of the completion. The shorter the prefix, the higher the compression ratio, and subsequently, the stronger the memorization. The authors further leverage an optimization-based approach using GCG, searching over the entire sequence length to find the shortest possible prefix, and demonstrate the effectiveness of such an approach in finding prompts which bypass in-context unlearning on LLaMA2-7-Chat & famous quotes.The authors perform further experiments on unlearning benchmarks such as TOFU and trying to forget harry potter, showing that ACR discovers that models are still able to reproduce significant portions of these datasets even after unlearning was applied. The authors also perform a scale-based analsyis as well as a comparison to data unlikely to be memorized to confirm that memorization increases with scale and the robustness of their proposed method, respectively.

**Strengths:**

- The paper is well written and positioned. It provides an excellent overview of the field of unlearning and motivates the contribution well.
- The experimental setup is exhaustive and well motivated.
- The proposed metric is demonstrably able to detect content which is still memorized within the model despite unlearning

**Weaknesses:**

- The optimization process, as mentioned by the authors, does not necessarily have to be completely accurate, sometimes underestimating the error ratio. It would be interesting to see how large this error can be by pehaps comparing the budget given to GCG. While it is not necessary for a solution to be optimal, an idea of how close the algorithm is, on average across a small sample size, would be relevant.
- The method identifies exact memorization, but does not account for paraphrases. While it is unlikely that a paraphrase would be memorized & easily reproduced, while the exact string forgotten, this might be a consequence of unlearning. What do you think about this issue?

**Questions:**

See above

**Limitations:**

Yes

---

> ### Author Rebuttal · Authors · 2024-08-07
>
> Thank you for your comprehensive review and valuable feedback.
>
> We appreciate your recognition of the strengths in our paper, particularly the well-positioned and motivated overview, exhaustive experimental setup, and the efficacy of our proposed metric in detecting memorized content despite unlearning efforts.
>
> Regarding the weaknesses you identified, we conducted additional experiments to address your points.
>
> **Limited budget for GCG**
> The idea is interesting, however, we found that controlling the budget does not significantly change things because our code is designed to exit early once a successful prompt (one that elicits an exact match) is found. Therefore, increasing the command line argument for num_steps does not provide additional benefits, as GCG is already optimized to use the necessary compute efficiently. And decreasing the number of steps shows many samples for which GCG fails to find a successful prompt altogether.
>
> **Paraphrases**
> We paraphrased the 100 famous quotes used in our paper with ChatGPT and checked their ACR values. The results are as follows:
>
> |Model	|Data Type	|Avg. ACR	|Portion Memorized|
> |:----------|:-----------------------------------|:--------|:------------|
> |EleutherAI/pythia-1.4b	|Paraphrase	|0.68	|0.11|
> |EleutherAI/pythia-1.4b	|Famous Quotes	|1.17	|0.47|
>
> We conducted a cursory Google search and found that some paraphrased versions are present on the internet. Our findings suggest that paraphrases of memorized content generally do not get flagged by our method unless the paraphrase itself is memorized. This supports the idea that "facts are not copyrightable," and our method aligns with this principle.
>
> Thank you again for your insightful comments. We believe these additions and clarifications will strengthen our paper.

---

> > ### Comment · Reviewer_kzBJ · 2024-08-13
> >
> > Thank you for your response and the additional results.
> >
> > Budget then is probably not a good way of quantifying this - nevertheless, some bounds on the error ratio/idea of proximity to the optimal solution seem necessary, especially with the intended purpose of the method in mind (quantifying memorization).
> > Example scenarios are: how much can the ACR vary between different optimization algorithms? How likely is the chance of the compression ratio seeming low, while the model actually memorized the texts?
> > Such scenarios are likely to pop up if ever the method is to be used as proof for memorization. This is definitely not an easy problem to solve, but should be kept in mind or discussed by the authors.
> >
> > My score (Accept, 7) still reflects my opinion of the paper well, thus I will keep it.

---

### Official Review · Reviewer_Um3a · 2024-07-16

**Soundness:** 2
**Presentation:** 3
**Contribution:** 3
**Rating:** 6
**Confidence:** 3

**Summary:**

### Summary

- This paper proposes a new metric for measuring memorization in LLMs.
- Their proposed metric called Adversarial Compression Ratio (ACR) is capable of measuring memorization for arbitrary strings at a reasonably low compute.
- The definition of memorization, the authors propose in the paper is based on a compression arguments which goes something like this - "a phrase present in the training data is memorized if we can make the model reproduce the phrase using a prompt (much) shorter than the phrase itself."
- To operationalize this definition, the techinque requires finding the shortest adversarial input prompt that is optimized to produce the sentence under consideration as its output.
- The ratio of input to output tokens is defined as ACR.
- There are various ways to measure memorization
	- Discoverable Memorization
		- This was proposed by Carlini and essentially measures if a prefix elicits the suffix as the response.
			- This definition is very permissive since definition only tests for exact completions.
			- It's easy to evade by just tweaking a few tokens to avoid exact match
			- It also requires validation dataset to tweak generation hyperaparameters such as top-p, top-k etc.
	- Extractable Memorization
		- This definition defines extractable memorization as a string which elicits the string in the response.
			- Since this allows any arbitrary string as prompt, this definition is very restrictive as it allows the prompt to have the entire string in question. For e.g., "repeat the following string: X" -> X
	- Counterfactual Memorization
		- Measures the difference in model performance with a model trained with the example versus one trained without it
			- Since LLMs are expensive to train, this definition is quite impractical.
	- Membership Inference Tests which are commonly used to test if a model was trained on a particular example datapoint has the following problems
		- It's very restrictive. Akin to plagiarism, it is okay to read copyrighted books but copying is problematic
		- Have brittle evaluation
- Based on the ACR ratio, ACR(M, y) = |y|/|x| where |x| = argmin |x| s.t. M(x) = y
  the authors propose a notion of $\tau$-compressible memorization i.e. if ACR(M,y) > $\tau(y)$
- This metric can be aggregated over a dataset to report an average compression ratio or another metric called portion memorized which measures the proportion of data with MCR >  $1$
- GCG (Greedy Coordinate Gradient) is used to find the the adversarial prompt from earlier work which is a common algorithm.
- The authors also show that in context unlearning can fool completion but not this adversarial notion of compression

**Strengths:**

### Strengths


- This definition is consistent with common notions of memorization
	- Bigger models memorize more
	- Random sequences are not compressible (zero compressibility)
	- sentences from a data source which are not part of training data have zero compressibility
	- Famous quotes have the highest value for ACR

- The paper is well written and easy to follow. The background is thoroughly covered.

**Weaknesses:**

## Weakness

- The paper is well motivated and tackles an important problem. However, independent of the problem of false negatives, I wonder if false positives might be a bigger problem here. What if as a result of GCG algorithm, you are able to elicit a generation from a model which was never seen by the model during the training. I would suggest adding an experiment and some discussion around this.

**Questions:**

## Questions

- Line 125: "Hard to arbitrate: training data is often not released."
	- Even if the training data is not released, can a publisher not run MIA against the LLM to determine effectively if their training data was used during training. I don't quite agree with this claim
- Line 313: Can you clarify what you mean "quells any fears that GCG is merely relaying that the gradients are more informative on some examples than others"

**Limitations:**

The authors have adequately addressed the limitations.

---

> ### Author Rebuttal · Authors · 2024-08-06
>
> Thank you for your thorough and thoughtful review. We are pleased that you found our definition of memorization consistent with common notions and that you appreciated the clarity and depth of our paper. We acknowledge your concerns and attempt to address them below:
>
> ### Re: False Positives
>
> We understand the importance of considering false positives in our method. Here are a few points addressing this concern:
>
> 1. **False Positives with GCG Algorithm**: The concern about the GCG algorithm eliciting generations that were never seen by the model during training is valid. Our experiments, including those involving random strings, have shown that while the GCG algorithm can elicit any generation, we have never observed compression of non-training data. This suggests that our method is robust against false positives. However, we will add further discussion and conduct an additional experiment to explicitly address this point. This experiment will aim to determine the likelihood of false positives by testing the ACR metric on a controlled set of non-training data.
>
> 2. **Reference to Related Work**: We will incorporate findings from relevant literature (e.g., Geiping, Jonas, et al. "Coercing LLMs to do and reveal (almost) anything." arXiv preprint arXiv:2402.14020 (2024)) to strengthen our discussion on false positives and demonstrate the robustness of our method.
>
> ### Re: Membership Inference Attacks (MIAs)
>
> We acknowledge the potential role of MIAs in determining training data usage. Here are our points addressing this issue:
>
> 1. **Limitations of MIAs**: While MIAs can be used to test for training data usage, they have significant limitations, as highlighted in recent work (e.g., https://arxiv.org/abs/2402.07841, https://arxiv.org/abs/2406.06443, https://arxiv.org/abs/2406.16201). Further, MIAs typically rely on scalar-valued losses, which are not easily interpretable in regulatory or legal contexts. This complicates their applicability in such settings and makes conclusive findings difficult to reach.
>
> 2. **Context of Our Claim**: Our claim in Line 125 refers to the broader challenges of proving training data usage without access to the training data itself. Even if a publisher can run MIA, the interpretability and conclusiveness of the results remain challenging. We will revise this section to provide more context and clarity on this point.
>
> ### Re: Clarity on Line 313
>
> We appreciate your request for clarification on Line 313. Here are our points addressing this query:
>
> 1. **Clarification on GCG and Gradient-Free Search**: One might think that our findings are the results of some peculiarity in GCG or some bias/preference GCG has for finding short prompts on some types of data. We establish that the same general trends in memorization can be observed with a gradient-free search algorithm, and thus conclude that we are not mistaking a GCG bias for some other signal. We will provide more details on this random search experiment and explain how it supports the robustness of our findings.
>
>
> Once again, we thank you for the constructive feedback on our work. Working on the pointers has helped us improve the quality of our analysis. We look forward to further discussions and improvements in this evolving field.

---

### Official Review · Reviewer_uS5V · 2024-07-16

**Soundness:** 2
**Presentation:** 4
**Contribution:** 2
**Rating:** 4
**Confidence:** 3

**Summary:**

This paper proposes Adversarial Compression Ratio (ACR), a metric for assessing memorisation in LLMs; ACR is defined as the ratio between the length of the generation we need to test memorisation for and the length of the shortest prompt that can elicit such a generation. A question I have is whether "length" is always suitable as a complexity measure; for example, y can be very long but also very "simple" (such as the "mmmmmmmm...mmm" messages from /r/microwavegang, which occur in some pre-training corpora). To solve such a combinatorial minimisation problem (finding the shorter prompt that elicits a given generation), the authors propose using Greefy Coordinate Gradient [GCG, Zou et al., 2023].

Overall, It is a very interesting approach and paper; however, it would be useful to compare ACR with other methods for testing for memorisation.

**Strengths:**

1) Interesting approach for memorisation testing

**Weaknesses:**

1) Lack of comparisons with other methods for solving the same task
2) Not sure why "number of tokens" may be a suitable measure of complexity

**Questions:**

Is ACR more effective at detecting memorisation compared to the method proposed in e.g., https://openreview.net/forum?id=TatRHT_1cK ?

**Limitations:**

Lack of comparisons with related methods for solving the same task

---

> ### Author Rebuttal · Authors · 2024-08-06
>
> Thank you for your insightful feedback. We appreciate your recognition of our innovative approach to memorization testing. We acknowledge your concerns and attempt to address them below:
>
> ### Re: Comparisons with Other Methods
>
> We understand the importance of comparing our Adversarial Compression Ratio (ACR) with other existing methods for testing memorization. Here are a few points addressing this concern:
>
> 1. **Verbal and Experimental Comparisons**: In Section 2 of our paper, we verbally compare and contrast ACR with existing methods. Additionally, our section on in-context unlearning includes some experimental comparisons. However, direct experimental comparisons are challenging due to the different contexts in which these methods operate. For instance, methods like Extraction (https://openreview.net/forum?id=TatRHT_1cK) can't make conclusive claims about short strings, whereas we can handle strings of any length. Also, the adversarial nature of our approach, where the model owner might take minor steps to change the extractability, also differentiates our method. The in-context unlearning section serves as an example where other tests might be tricked, but our approach remains robust. More concretely, a direct comparison with the paper requested can be found in Section 4.1 of the paper, and also in Section 4.2 (Figure 3) on compression versus completion. These results clearly demonstrate how compression is a desirable metric to uncover seemingly hidden memorization.
>
> 2. **Effectiveness of ACR**: ACR introduces a new paradigm of memorization detection from an adversarial perspective. This approach has significant implications for discussions on copyright and intellectual property, where model providers may take measures to hide memorization. We believe that this perspective enriches the discourse and offers new insights for policy-making and regulation. It is more effective in the sense that it pierces the illusion of compliance, showing memorization where other methods might fail to, for example because of system prompts.
>
> ### Re: Suitability of "Number of Tokens" as a Complexity Measure
>
> We acknowledge the concern regarding the use of "number of tokens" as a measure of complexity. Here are a few points addressing this concern:
>
> 1. **Compression Ratio**: The compression ratio, where both the numerator (length of the generation) and the denominator (length of the shortest prompt) must be in the same units, is helpful in enforcing the constraint that the prompt has less information than the output. While other notions of information content could theoretically be used, we find the ratio of token lengths to be practically useful in defining memorization.
>
> 2. **Empirical Utility**: Empirically, we have found that using the number of tokens as a measure of complexity works effectively in our experiments. The simplicity of this metric allows for a clear and consistent definition of memorization across different contexts.
>
>
>
> Once again, we thank you for the constructive feedback on our work. We hope we were able to clarify your concerns. Please let us know if there are any lingering concerns.

---

### Author Rebuttal · Authors · 2024-08-07

We appreciate the thoughtful feedback and valuable suggestions from all of the reviewers.

In response, we provided further clarity around key assumptions, conducted some experiments, and expanded our discussion in several places in the draft. Specifically, we examined paraphrased versions of the famous quotes and found that our methods align well with copyright standards. We believe all of this improvement have strengthened our paper and and that we have comprehensively addressed the reviewers' concerns.

One reviewer asked *How does the ACR metric account for variations in the length of expressions?*
Here, we include a plot of the average ACR versus sample length in the attached PDF (and in the latest version of the paper). We find no apparent spurious correlations, but rather that our method is useful across a range of sample lengths. For further accounting for information content of the samples, Appendix E.2 in the paper has a discussion of how the ACR compares to other compressibility measures like SMAZ. In conclusion, we agree that the concern about possible correlations with sample length should be addressed and we have added a plot and some discussion to better explain that this is not an issue for our method.

---

### Author Response · Authors · 2024-08-13

Dear reviewers,

As we approach the end of the discussion period, we kindly inquire whether we have addressed your concerns raised in the reviews. We appreciate your feedback and any further questions. They greatly help us improve our work. Thanks for your time and careful reviews.

Best,
All authors

---

### Comment · Senior_Area_Chairs · 2024-08-13

Dear reviewers,

The discussion period will end soon. If you haven't responded to the authors' rebuttal, please do so and kick off the discussion.

Best,
SAC

---

### Decision · Program_Chairs · 2024-09-25

**Decision:**

Accept (poster)

**Comment:**

This paper proposes an interesting new definition for what constitutes as memorization in LLMs -- if there exists a prompt which elicits the target text and is shorter than that text, then we can say the model has memorized the target. This makes intuitive sense and the paper does a great job of contrasting this method with other popular notions of memorization.

The main weaknesses, as noted by the reviewers, are around the empirical evaluation of the method:
1. There is a lack of direct comparison to other methods for detecting memorization, such as discoverable and extractable memorization. The authors rightfully claim that this is difficult as the definition of memorization itself is different across these methods, but at the very least it would have been instructive to see a comparison of *how many* sequences from the training data each method can labels as memorized.
2. The issue of false positives is not adequately addressed in the paper. There are many sequences to which LMs assign a high probability despite having never seen them during training. An experiment showing how compression ratio changes with perplexity on unseen sequences would be instructive in determining the validity of the method.

Despite the above limitations, overall the paper is interesting and tackles an important problem. Hence, I am recommending a weak accept -- with the hope that the authors will address the points above for the camera ready version.